# Influence of Ni and Sn Perovskite NiSn(OH)_6_ Nanoparticles on Energy Storage Applications

**DOI:** 10.3390/nano13091523

**Published:** 2023-04-30

**Authors:** G. Velmurugan, R. Ganapathi Raman, D. Prakash, Ikhyun Kim, Jhelai Sahadevan, P. Sivaprakash

**Affiliations:** 1Department of Physics, Noorul Islam Centre for Higher Education, Kumaracoil, Kanyakumari 629180, Tamil Nadu, India; velgi_81@yahoo.com; 2Department of Physics, Saveetha Engineering College (Autonomous), Chennai 602105, Tamil Nadu, India; 3Department of Physics, Kongunadu College of Engineering and Technology, Thottiyam 621215, Tamil Nadu, India; duraiprakas@gmail.com; 4Department of Mechanical Engineering, Keimyung University, Daegu 42601, Republic of Korea; siva.siva820@gmail.com; 5Centre for Material Science, Department of Physics, Karpagam Academy of Higher Education, Coimbatore 641021, Tamil Nadu, India; jhelaidev@gmail.com

**Keywords:** Ni, Sn perovskite hydroxide, energy density, asymmetric supercapacitor, power density, electrochemical stability, cyclic voltammetry

## Abstract

New NiSn(OH)_6_ hexahydroxide nanoparticles were synthesised through a co-precipitation method using various concentrations of Ni^2+^ and Sn^4+^ ions (e.g., 1:0, 0:1, 1:2, 1:1, and 2:1; namely, N, S, NS-3, NS-2, and NS-1) with an ammonia solution. The perovskite NiSn(OH)_6_ was confirmed from powder X-ray diffraction and molecule interactions due to different binding environments of Ni, Sn, O, and water molecules observed from an FT-IR analysis. An electronic transition was detected from tin (Sn 3d) and nickel (Ni 2p) to oxygen (O 2p) from UV-Vis/IR spectroscopy. Photo luminescence spectroscopy (PL) identified that the emission observed at 400–800 nm in the visible region was caused by oxygen vacancies due to various oxidation states of Ni and Sn metals. A spherical nanoparticle morphology was observed from FE-SEM; this was due to the combination of Ni^2+^ and Sn^4+^ increasing the size and porosity of the nanoparticle. The elemental (Ni and Sn) distribution and binding energy of the nanoparticle were confirmed by EDAX and XPS analyses. Among the prepared various nanoparticles, NS-2 showed a maximum specific capacitance of 607 Fg^−1^ at 1 Ag^−1^ and 56% capacitance retention (338 Fg^−1^ and 5 Ag^−1^), even when increasing the current density five times, and excellent cycle stability due to combining Ni^2+^ with Sn^4+^, which improved the ionic and electrical conductivity. EIS provided evidence for NS-2’s low charge transfer resistance compared with other prepared samples. Moreover, the NS-2//AC (activated carbon) asymmetric supercapacitor exhibited the highest energy density and high-power density along with excellent cycle stability, making it the ideal material for real-time applications.

## 1. Introduction

Supercapacitors are the most demanding environment-sustainable energy storage devices due to their superior capacitive nature, high-power density, and protracted charge/discharge cyclic stability [1]. Supercapacitors or ultra-capacitors are alternatives to fossil fuels, which helps to reduce air pollution. In order to address energy issues, renewable energy sources such as solar and wind have been studied recently [2], although an admittance of these assets is not constantly accessible. Thus, emerging pertinent energy storage devices are critical to large storage abilities. Supercapacitors are suitable devices to store electrical energy, and can be used as power backup systems, handy electronics, E-vehicles, and in other applications [3]. Supercapacitors have been extensively reported with decent performance. Improving the capacitance of the supercapacitor’s performance depends on the electrochemical performance of the electrode material [4]. Metal fluoride/hydroxides/oxides/telluride (e.g., NiF_2_, SnO(OH)_2_, V_2_O_5_, Ni(OH)_2_, Co(OH)_2_, MnO_2_, NiO, SnO_2_, Co_3_O_4_, and MnFe_2_O_4_) were used as pseudocapacitive materials for the first few decades on interpretations of their greater specific capacitance and energy density [5,6,7,8,9].

Particularly, SnO_2_ and SnO_2_–carbon nanostructures were used for various applications such as electrochemical biosensors, optical limiting, piezoelectric sensors, batteries, and supercapacitor applications [10,11,12]. On the other hand, metal binary hydroxides are attractive due to their superior electrochemical properties, which have been addressed in the literature; Ni-Mn hydroxide and Ni-Co binary hydroxide, with improved pseudocapacitive behaviours, were reported [13,14]. Here, Ni was substituted with a tin precursor to enhance the supercapacitor’s performance. Hence, nickel hydroxide (Ni(OH)_2_) and tin oxy-hydroxide (Sn_6_O_4_(OH)_4_) are suitable pseudocapacitive double-hydroxide materials for energy applications. Ni(OH)_2_ with tin hydroxide has a variety of gains, including a low cost, less toxicity, and being eco-friendly, which grades it as a noble option for energy storage applications. The formation of the octahedral hexahydroxostannate (IV) anion [Sn(OH)_6_]^2−^ dominates the formation of tetravalent tin ternary hydroxides and attests to this anion’s high chemical stability. From this double metal, we now propose a new matrix based on hydroxide to promote the redox property of Ni- and Sn-based perovskite hydroxide.

Herein, a co-precipitation process was implemented to synthesise NiSn(OH)_6_ containing a double-metal hexahydroxide electrode for the supercapacitor as an electrochemical energy storage application. The electrochemical capacitance of NiSn(OH)_6_ was systematically studied with the effect of various concentrations of Ni^2+^ and Sn^4+^. This material patron’s high ion diffusion enhanced the specific capacitance of the materials. This approach delivered a suitable composition, and a controlled identical particle size could be obtained through homogeneity, which was confirmed by FE-SEM, EDAX, and XPS. Of the prepared nanoparticles, NS-2 had the highest specific capacitance of 607 Fg^−1^ at 1 Ag^−1^, admirable capacitance retention (56% (338 Fg^−1^) and 5 Ag^−1^), and excellent cycle stability due to the combination of Ni^2+^ and Sn^4+^, which improved ionic and electrical conductivity.

Moreover, NS-2 had more oxygen defects, confirmed by second-order Raman modes obtained at 1100 cm^−1^, which influenced and provided more active sites for redox reaction, leading to the highest specific capacitance. Furthermore, the NS-2//AC asymmetric supercapacitor had the highest energy density, the highest power density, and the best cycle stability. NiSn(OH)_6_ is an excellent candidate for real-time supercapacitor applications.

## 2. Experimental and Synthesis Techniques

### 2.1. Materials

Chemicals, including nickel chloride (NiCl_2_·2H_2_O), tin chloride (SnCl_2_·6H_2_O), and an ammonia solution, used in this investigation were procured from Sigma−Aldrich (99.99%) [Oakville, Canada].

### 2.2. Synthesis of NiSn(OH)_6_

Ni*_x-y_*Sn*_y_*(OH)_6_ was synthesised using the co-precipitation method. In total, 1 M NiCl_2_·2H_2_O and different molecular ratios (0, 2 M, 1 M, and 0.5 M) of tin chloride were dissolved in 80 mL of de-ionised water and stirred for 2 h at 80 °C. Hydrochloric acid was added dropwise into the solution and the solution reached a pH value of 0.7, which can meritoriously inhibit the hydrolysis of Sn^2+^ and Ni^2+^. The ammonia solution was added dropwise to the above mixture until the pH reached 10. As the pH increased from 0.7 to 12.0, a pale green sol–gel product appeared in the solution, which could be expressed by the reaction in Equation (1). After precipitation formation, the NiSn(OH)_6_ product was centrifuged and dried on a hot plate at 90 °C for 12 h. The schematic representation of the NiSn(OH)_6_ sol–gel occurred from the co-precipitation method, as shown in Figure 1.
Ni^2+^ + Sn^4+^ + 6OH^−^ = NiSn(OH)_6_(1)

### 2.3. Electrode Fabrication and Capacitance Calculations

Polyvinylidene fluoride (PVDF), acetylene black (AB), and an electroactive material were mixed at a mass ratio of 80:10:10 in N-methyl-2-pyrrolidone to produce the working electrode (NMP). About 1 mg of homogenous slurry (including the mass of PVDF and acetylene black) was coated on an area of 1 cm × 1 cm on the 1 cm × 3 cm nickel foam used as the current collector. The coated foam was dried at 90 °C overnight for solvent evaporation. The electrochemical studies were carried out in three-electrode configurations with a supporting electrolyte of 6 M KOH. KCl-filled Ag/AgCl and platinum wires were used as the reference and counter electrodes, respectively. The following equations were used to calculate the active material’s specific capacitances in the three-electrode configuration as well as the CV and GCD curves [15,16].

From the CV curves, the specific capacitance active material was
(2)$$C_{sp}=\int \frac{IdV}{vmV}$$

From the GCD curves, the specific capacitance active material was
(3)$$C_{sp}=\frac{Idt}{mdV}$$

The asymmetric supercapacitor (ASC) specific capacity was improved by adjusting the mass ratio of the negative electrode (activated carbon (AC)) and positive electrode (NS-2 (NiSn(OH)_6_)) using the following Equation (4) [15,16]:(4)$$\frac{m^{+}}{m^{-}}=\frac{C^{+}}{C^{-}}\times \frac{{∆V}^{+}}{{∆V}^{-}}$$
where m^+^ and m^−^ are the mass of the negative and positive electrodes, respectively; ∆V^+^ and ∆V^−^ are the potential windows; and *C*^+^ and *C*^−^ are the specific capacitance for the positive and negative electrodes, respectively, in the three electrodes. For the ASC, the mass ratio was taken as 1:4 of NS-2 (NiSn(OH)_6_) and AC, including the mass of acetylene black and PVDF. The energy and power density of the ASC were calculated using the following equations [15,16].
(5)$$E=\frac{{C\;\Delta V}^{2}}{2\times 3.6}$$
(6)$$P=\frac{E\;\times \;3600}{\Delta t}$$

### 2.4. Characterisation

The powder XRD technique was used to discover the crystal structure of the prepared samples in the range of a 10–80° angle (Model: Rigaku Corporation, Tokyo, Japan). Further, FE-SEM was performed using a ZEISS ultra 55 microscopes to record the microstructure of the materials, and Raman spectroscopy was performed using a Renishaw appliance (Renishaw Invia) with a 532 nm green laser wavelength. In order to assess the sample surface’s absorption, UV-DRS was performed using a Holmarc instrument (Holmarc Opto-mechatronics, Kochi, India, Model: Ho -SP-DRS 100). To determine the electronic structure of the materials, the photoluminescence spectrum was measured using Horiba equipment. Furthermore, a Biologic SP300 was used for the electrochemical analysis of NiSn(OH)_6_ in the supercapacitor applications.

## 3. Results and Discussions

### 3.1. X-ray Diffraction (XRD)

The various ratios of Ni and Sn hydroxide powders were prepared through co-precipitation method preparations, as illustrated in Figure 2. Powder XRD patterns of the as-prepared Ni(OH)_2_, SnO(OH)_2_ NS-1 (NiSn_2_(OH)_6_), NS-2 (NiSn(OH)_6_), and NS-3 (Ni_2_Sn(OH)_6_) are shown in Figure 2. The diffractions at the 2θ values of 19.8°, 33.4°, 38.8°, 52.6°, and 62.9° were indexed to (001), (100), (101), (102), (110), (111), (103), and (201) planes, respectively, for the hexagonal phase and were well matched with the JCPDS: 14-0117 of Ni(OH)_2_, confirming the presence of β-Ni(OH)_2_ [17]. Diffraction peaks at the 2θ values of 19.3°, 31.7°, 37.1°, 45.1°, 59.7°, and 65.6° shown in Figure 2 were attributed to Sn_6_O_4_(OH)_4_ and the corresponding JCPDF #01-084-2157 of the S sample. NS-2 clearly showed that diffraction peaks at 11.14°, 19.35°, 38.49°, and 51.92° corresponded with the planes (003), (001), (101), and (102), and shifting the peaks confirmed the substitution of Ni in Sn hydroxide. With Ni and Sn concentration ratios of 1:0, 1:2, 1:1, and 2:1, there were broad intensity diffraction peaks, except for 0:1 (Sn_6_O_4_(OH)_4_), due to the occurrence of hydrogen bonding between Ni-OH and Sn-OH in NiSn(OH)_6_ as well as a decrease in crystallite size [18]. The 2θ shifts were also due to Sn and Ni ion rearrangements and quick crystallisation; atoms reformed in the crystal lattice, which occurs in an interaction between Ni-OH and Sn-OH. The diffraction peaks had a low intensity, which indicated the presence of nanocrystalline materials. This is good for supercapacitor applications for the reason that ions could easily permeate through the bulk of the NiSn(OH)_6_ material.

### 3.2. Fourier Transform Infrared Spectroscopy

The FT-IR spectra of samples N, S, NS-1, NS-2, and NS-3 (NiSn(OH)_6_) are shown in Figure 3. The sharp peak at 3638 cm^−1^ was due to the non-allowance of water molecules into the space of the inner-layered Ni(OH)_2_ and to an arrangement of hydrogen bonds for the confirmation N sample [19]. Broad peaks for the other sample (NiSn(OH)_6_) at ~3455 cm^−1^ designated the O-H group’s stretching mode. The peak at 1619 cm^−1^ reflected the O-H group’s bending vibrations in Sn_6_O_4_(OH)_4_ and Ni(OH)_2_. The bands at 529 and 425 cm^−1^ designated the in-plane deformation or bending vibration of the OH^−^ group, and the stretching vibrations of Ni–O suggested that Ni(OH)_2_ was in the β-phase. The typical Sn-O bond-stretching vibrations at 522 cm^−1^ and 1006 cm^−1^ could be ascribed to the formation of Sn_6_O_4_(OH)_4_ [20]. The vibration bands at 625 cm^−1^, 509 cm^−1^, and 445 cm^−1^ with different ratios of Ni and Sn in the Ni_x-y_Sn_y_(OH)_6_ stretching vibrations of Ni–O and Sn-O inferred the combination of Sn ions within the Ni(OH)_2_ lattice.

### 3.3. UV-Vis Spectra and PL Spectra for Optical Studies

The absorption spectra of Ni(OH)_2_, SnO(OH)_2_, NS-1, NS-2, and NS-3 (NiSn(OH)_6_) showed (Appendix A) a peak in the range of 288–307 nm, which was attributed to an electron transition from tin (Sn 3d) and nickel (Ni 2p) to oxygen (O_2p_). The absorption spectra redshifts in NS-1, NS-2, and NS-3 (NiSn(OH)_6_) with respect to pure (N) Ni(OH)_2_ (284 nm) and S (Sn_6_O_4_(OH)_6_) indicated the inherent electronic structure of Ni and Sn [21]. This modification was caused by a local disorder and s and p-d hybridisation. Figure 4 shows the photoluminescence spectra of N, S, NS-1, NS-2, and NS-3 at room temperature for an excitation wavelength of 275 nm. The emitted energies of the UV region from PL were 3.23, 3.18, 3.12, and 3.14 eV for N, S, NS-1, NS-2, and NS-3, respectively. The PL spectra were redrawn, baseline adjusted, deconvoluted, and fitted with a Gaussian line shape. The PL spectral lines of all samples in the entire spectrum range could be fitted using seven components for the N sample, five components for the S sample, and six components for the remaining samples. Sample N exhibited seven PL bands in the UV and violet-blue regions whilst the other samples exhibited blue-green emissions. When we increased the doping, the UV emission slightly diminished and blue-green emission monotonously peaked. Thus, the fitting results showed that the PL spectra could be described by 5, 6, and 7 PL components, which demonstrated the different trends with increasing the doping of NP concentrations. The visible region had three shoulder peaks centred at 411–425, 442–447, and 484–492 nm. The Figure 4 depicts a potential emission mechanism for N, S, NS-1, NS-2, and NS-3. As a result, the energy emitted from the UV region was ascribed to the near-band edge emission (NBE) [22,23]. As a whole, emissions of 400–800 nm in the visible region of semiconducting oxide materials were caused by oxygen vacancies due to various oxidation states (doubly ionised, singly ionised, four-ionised, and a neutral form) of Ni and Sn metals.

### 3.4. Raman Spectroscopy

Raman spectroscopy can be used to determine the structural defects, crystallinity, size, and dopant distribution influence of nanoparticles. The Raman spectra of the prepared Ni(OH)_2_, SnO(OH)_2_, NS-1, NS-2, and NS-3 (NiSn(OH)_6_) are shown in Figure 5. The group theory believes that Raman-active transitions E_g_ at 582 cm^−1^ are based on the symmetry of Ni(OH)_2_. A_1g_, A_2g_, A_2u_, B_1g_, B_2g_, 2B_1u_, E_g_, and 3E_u_ are normal vibrational modes of SnO_2_ [24]. Raman-active modes are A_1g_, B_1g_, B_2g_ (non-degenerate), and E_g_ (doubly degenerate). There are two active IR modes: A_2u_ (singlet ongitudinal optical (LO)) and E_u_ (triply degenerate: transverse optical (TO)). A_2g_ and B_1u_ are the silent modes. The Eg mode is ascribed to the oxygen vibrations and the corresponding peak at ~467 cm^−1^. The expansion–contraction of the Sn-O bond is indicated by the B_2g_ (769 cm^−1^) and A_1g_ (~630 cm^−1^) modes. The maximum number of hydroxyl groups Ni and Sn present in a solution increases the nucleation process, resulting in local disorders (defects) and oxygen vacancies. Almost all the modes (Eu (2) TO, E_u_ (2) LO, E_u_ (3) TO, B_1u_ (3), A_2u_ TO, and A2u LO) were observed in NS-2 and other samples, and were represented by the bands at ~247, ~305, ~340, ~505, ~542, and ~692 cm^−1^, respectively [24]. The second-order modes were also observed at 1100 cm^−1^, which is typically very weak, and intensified disorder in NS-2, which was reliable with the reported correlation of intensity against crystallinity.

Field-emission scanning electron micrographs (FE-SEM) inspected the morphology of the prepared samples. The Figure 6a,b magnification of the FE-SEM image implied that Ni(OH)_2_ and Sn_6_O_4_(OH)_4_ had a particle nature. Further, the double-hydroxide porous materials with increased particles size formed a spherical shape by modifying the Ni and Sn molecular concentration of NS-1 for (1:2), NS-2 for (1:2), and NS-3 for (2:1); the similar structure is shown in Figure 6c–e. Compared with NS-1 and NS-3, NS-2 had a uniform particle size and porous structure, which is helpful for electrochemical energy storage applications. Further, the elemental composition of the prepared samples was premeditated. The energy-dispersive X-ray photoelectron spectrum analysis shown in Figure 6f and Appendix A complied with XPS. The elemental composition of the prepared samples is presented in Table 1.

### 3.5. XPS Analysis

The elemental composition with a valence state analysis of N, S, NS-1, NS-2, and NS-3 (NiSn(OH)_6_) was confirmed by the XPS analysis, and is shown in Figure 7a–d. The XPS survey spectra (Figure 7a) and the existence of Ni, Sn, and O species were observed absolutely. Auger peaks ascend from Sn and Ni, and exist in the spectrum. Here, they delivered information about the oxygen atom’s binding energy and disclosed the composition of oxygen and the oxygen vacancy in the samples. Due to the very small particle size of NiSn hexahydroxides, they have superior oxygen vacancies, which helps in electrochemical energy storage applications. Figure 7a shows a spectrum of N, S, and NS-2 samples, endorsing the occurrence of Ni, Sn, and O elements. Binding energies of ~493.9 eV and ~485.4 eV for Sn 3d_3/2_ and Sn 3d_5/2_, respectively, confirmed the Sn^4+^ oxidation state [24,25]. These peaks for NS-2 shifted to higher binding energies (~486.8 and ~495.4 eV) due to the Ni ions in the tin hydroxide lattice forming NiSn(OH)_6_. Figure 7b depicts the Ni 2p deconvoluted spectrum (NS-2) with doublet Ni 2p_3/2_ and Ni 2p_1/2_. The Ni 2p_3/2_ constituent was committed into four peaks: 855.2, 856.6, 861.7, and 866.1 eV. The peak at 856.6 eV was the primary peak and the other three were satellite peaks; these were observed at 855.2, 861.7, and 866.1 eV [26,27]. The occurrence of Ni^2+^ vacancies and Ni^3+^ ions in Ni^2+^-OH species has been associated with the peak at 856.6 eV, although the satellite peak at 861.7 eV comprises a charge transfer between Ni and Sn. Therefore, the Ni atoms in the NiSn(OH)_6_ electrode had a different native environment, designating the existence of Ni and Sn in the samples [28,29,30,31,32]. The Ni^2+^ and Ni^3+^ oxidation states had binding energies of 854.9 and 856.5 eV, respectively. The occurrence of both Ni^2+^ and Ni^3+^ in the NS-2 samples correlated with the NiSn(OH)_6_ structure from XRD. The satellite peaks were responsible for the other two binding energies, 861.7 and 866.6 eV. Further, the Ni 2p_1/2_ component peak at 874.8 eV represented the Ni^3+^ oxidation state of Ni ions. The deconvolution of Sn exactly matched the raw data-binding energies of ~493.9 eV and ~485.4 for Sn 3d_3/2_ and Sn 3d_5/2_ (Figure 7c), respectively, confirming the Sn^4+^ oxidation state [26,27]. The deconvolution O 1s spectrum of NS-2 (Figure 7d) had a peak at 531.4 eV representing OH^−^ ions and oxygen vacancies. Generally, the major peak at 530.9 eV was typically observed in Sn-O for SnO_2_. The peak shifted a higher binding energy of 531.4 eV, which is predominantly associated with the occurrence of ≡Sn-OH species. On the other hand, the peak shift may have been due to O atoms bonding with Ni atoms, which arises from certain Sn^4+^ atoms being replaced by Ni^2+^ in the Sn(OH)_4_ lattice [25,28,29,30]. In Figure 7c,d, the binding energies inferred the formation of bigger sphere-like particles of NS-1, NS-2, and NS-3 compared with bare N (Ni(OH)_2_) and Sn_6_O_4_(OH)_4_.

### 3.6. Electrochemical Studies

#### 3.6.1. Cyclic Voltammetry Studies

The electrochemical performance of prepared samples of N (Ni(OH)_2_), S (Sn_6_O_4_(OH)_4_), NS-1, NS-2, and NS-3 (NiSn(OH)_6_) in a three-electrode configuration was investigated using CV and GCD, with 6 M of KOH as the supporting electrolyte. Figure 8 depicts that the CV curves of the prepared sample had a potential range (0 to 0.6 V) with scan rates of 10 mV/s. The redox peak in CV represented the composite’s faradic behaviour and was caused by variations in Sn and Ni oxidation states. The CV profile enclosed integral area and current density of NS-2 were greater than those of the other samples, indicating that it had maximum capacitance. Oxidation and reduction peaks were observed at potential (0.35 V) and (0.30 V) scan rates of 10 mV/s, respectively. The highly performing candidate’s electrochemical capacitive mechanism could be due to porous particle morphology and oxygen vacancies. This porous particle morphology with less agglomeration had a higher conductivity, and would be more sensitive to electrolyte ions during redox reactions. Due to its porous formation, it may also withstand with its robustness and capacitive behaviour throughout the reaction better than other electrodes, whereas other electrodes may experience clumsiness at the electrode/electrolyte interface, which blocks the active sites. Oxygen vacancies are important in improving conductivity and decreasing diffusion energy, which could synergistically improve capacitive behaviour. The proposed faradic redox reaction mechanism of the prepared samples is given below.
NiSn(OH)_6_ ↔ NiSn(OOH)_6_ + 6H_2_O + 6e^−^

Figure 8b–f show the cyclic voltammetry features of N, S, NS-1, NS-2, and NS-3 (NiSn(OH)_6_) with potentials ranging from 0 to 0.6 V and scan rates of 10, 20, 50, and 100 mV/s, respectively. By increasing the scan rates, the area under CV curve also increased, which indicated the high-power characteristics of the prepared nanoparticle. The oxidation and reduction peaks moved to the positive and negative; i.e., increasing and decreasing the cathodic and anodic potentials, respectively. Oxidation and reduction peaks were gradually shifted for the potential (0.35–0.5 V) and (0.35–0.25 V) scan rates from 10 to 100 mV/s, respectively. The current increases caused by the increasing scan rate as well as the CV peak shift in the higher potential could be ascribed to the electrode’s relaxed inherent ohmic limitation and electrochemical kinetics [29,31].

#### 3.6.2. Galvanostatic Charge/Discharge Method

The charge storage nature of the electrode was revealed by galvanic charge/discharge (GCD) measurements at constant current values with the potential time frame. The Figure 9 depicts the charge/discharge behaviours of the prepared samples over a potential range of 0 to 0.6 V at 1 A/g. Figure 9a shows that the GCD curves of all samples exhibited excellent electrochemical reversibility. All the prepared samples displayed pseudocapacitive behaviour in the non-linear GCD profile. The electrochemical redox reaction that resulted confirmed the faradaic charge storage characteristics of all samples. The different discharging times of the prepared samples of 162.5–273.5 s demonstrated the admirable pseudocapacitive nature of NiSn(OH)_6_. This result agreed well with the results of the CV curves. The incompletely reversible redox reaction caused an asymmetry in the GCD curves. Figure 9a shows a negligible IR drop, indicating a low corresponding series resistance, which has an impact on the whole performance of a supercapacitor [29,31,32]. Equation (2) was used to calculate the specific capacitance from the GCD.

As demonstrated, the specific capacitances for N, S, NS-1 (2:1), NS-2 (1:1), and NS-3 (1:2) were significantly affected by the different molecular weights of Sn and Ni. The products of equal concentration of Sn, Ni, and NS-2 had the highest specific capacitance of 607 Fg^−1^, followed by the products of NS-3 (565 Fg^−1^), NS-1 (509 Fg^−1^), N (380) Fg^−1^, and S (323 Fg^−1^) (Appendix A). The highest specific capacitance for the composite prepared with (1:1) MW Sn and Ni matched the CV results. Furthermore, the NS-2 product contained an equal amount of Sn and Ni, resulting in decreased resistance and more active NiSn(OH)_6_ sites. According to the redox mechanism of NiSn(OH)_6_, it reacted with the KOH solution, which could offer more active sites to enhance the capacitance [29,31].

Figure 9c depicts the electrochemical properties of the N, S, NS-1, NS-2, and NS-3 (perovskite NiSn(OH)_6_) prepared electrodes with a different molecular ratio of Sn and Ni at various current densities. The specific capacitances calculated from GCD curves were much higher than 607 Fg^−1^ at current densities of 1 Ag^−1^. When the current density was increased to 5 Ag^−1^, the specific capacitances gradually decreased to 338 Fg^−1^ (Figure 9c). Similarly, other samples also decreased their specific capacitance with an increasing current density, and the values are shown in Table 2. This could have been due to insufficient time for redox reactions. When the current density is lower, there is more time for ions to diffuse into the active materials of the electrode, allowing for an adequate interaction between an electrolyte and an electrode. However, higher current densities do not provide enough time to allow this interaction, which causes the capacitance to decrease whilst increasing the current density [6,8].

Figure 9d also shows the charge/discharge (CD) cycle stability and columbic efficiency of the prepared materials at 5 Ag^−1^. After 5000 charging and discharging cycles, the capacitance was still approximately 78% of its initial value of capacitance. After the first 1500 cycles, the capacitance stabilised and a tiny decrease in capacitance was observed, indicating high capacitance stability. The NS-2 NiSn(OH)_6_ electrode material changed due to the immersion of the electrolyte during the charge and discharge process. There was no decreasing tendency from the first cycles, demonstrating that no activation occurred over all the cycle stability processes, and the decrease could be improved by combining the two factors. This was due to a change in the inner microstructure by changing the method of making the electrodes or by using an aqua KOH electrolyte [6,8]. Alternatively, there was a strong correlation between the cyclic performance and the material nature such as microstructure crystallinity and porosity. The material’s crystallinity ensured a good cycle performance. The comparison of the specific capacitance values from the literature and the prepared samples is shown in Table 2.

As an open circuit potential, electrochemical impedance spectroscopy (EIS) was performed on N, S, NS-1, NS-2, and NS-3 (perovskite NiSn(OH)_6_) electrodes with 1 M Na_2_SO_4_. These measurements were used to compute important parameters such as the charge transfer resistance, solution resistances, electric double-layer capacitance (Cdl), Warburg element (Zw), and pseudocapacitance (Cp). Figure 10 depicts typical Nyquist plots for N, S, NS-1, NS-2, and NS-3 (perovskite NiSn(OH)_6_) electrodes. The Rs were found by intersecting the high-frequency side of the Nyquist plot with the real axis. In the high-to-medium frequency range, the Rct values of the electrodes were measured through the semicircle. A straight line with an inclination greater than 45° accompanying the low-frequency region represents the ideal capacitive behaviour of the electrodes representing the Warburg impedance. Zw could also be expressed as the electrodes’ OH^−^ diffusive resistance [15]. N, S, NS-1, NS-2, and NS-3 (perovskite NiSn(OH)_6_) had equivalent series resistances (ESRs) of 25.55, 23.27, 17.21, 7.29, and 11.57 ohm, respectively. The Rct value of the NS-2 electrode was lower than that of the N, S, NS-1, and NS-3 (perovskite NiSn(OH)_6_) electrodes. This R_ct_ value also confirmed that NS-2 was the best supercapacitor electrode. However, the focus of this paper was on the synthesis of an NS-2 (NiSn(OH)_6_) electrode material with a different structure and good electrochemical properties.

#### 3.6.3. ASC Device Two-Electrode Electrochemical Studies

The electrochemical study activity was revealed in NS-2. We easily fabricated an ASC with activated carbon (AC) used as a negative electrode with judicious energy and power density alongside our NS-2 used as a positive electrode, avoiding the use of expensive carbons (graphene, CNTs, etc.). The electrochemical performance of AC and NS-2 was distinctly investigated using CV at a scan rate of 10 mVs^−1^ in 6 M KOH (Figure 11a). The quasi-rectangular shape of CV cultured for AC, with no redox peaks, indicated that AC had EDLC properties, whereas the CV of NS-2 had oxidation and a strong reduction reaction as predicted, indicating pseudocapacitive behaviour [6]. The optimum voltage window was determined by carefully analysing the results obtained for both NS-2 and AC. The ASC’s charge/discharge process is illustrated below.
NiSn(OH)_6_ ↔ NiSn(OOH)_6_ + 6H_2_O + 6e^−^//6C + 6KOH ↔ 6CK^+^ + 6OH^−^

The optimal mass ratio for fabricating the ASC from the equation was determined by analysing the specific capacitance of NS-2 (NiSn(OH)_6_) and AC. As shown in Figure 10, the ASC device was exposed to a CV analysis in a PVA/KOH gel electrolyte at scan rates of 10, 20, and 50 mVs^−1^. The CV curves had a somewhat similar shape to that noticed from the three-electrode configurations of NS-2 (NiSn(OH)_6_). However, as the scan rate increased, the collective effect of the redox characteristics from NS-2 (NiSn(OH)_6_) and the EDLC properties from AC could be seen (Figure 11b), although NS-2 (NiSn(OH)_6_) showed a better capacitance observed within a 1.3 V potential window when the optimised mass ratio was used to fabricate the asymmetric capacitor.

The constructed ASC further evaluated the real values of energy and power density, which are two key factors for evaluating the performance of supercapacitors [30]. Figure 10c shows GCD plots at various current densities (1–5 A/g) with a potential window of 1.3 V. The CD curve was a triangular shape (Figure 11c) up to 5 Ag^−1^, demonstrating the good charge transport behaviour of electrolyte ions in the NS-2 and AC electrode. The capacitance of the ASC was calculated using Equation (6), and the values were 29 Fg^−1^, 17.5 Fg^−1^, 12 Fg^−1^, and 9.5 Fg^−1^. The Ragone plot for NS-2//AC ASC is shown in Figure 11d. The NS-2//AC sample ASC had an energy density of 2.5 to 7 Wh/kg and a power density of 685 to 1784 W/kg, which was a comparable value to double-metal oxides [31,32,33] or other Ni/Co hydroxides. These outcomes may help to explain the unique structure of NS-2. The porous particle structure benefits ion and electron diffusion routes, increasing the capacitance and rate capability. On the other hand, Ni^2+^ and Sn^4+^ have synergistic effects in NiSn hexahydroxide, which significantly contribute to the modification of their physical properties and the facilitation of electron-transfer processes.

Stability was observed, even after 2000 consecutive cycles of GCD at a current density of 5 Ag^−1^ between 0 and 1.3 V. The capacitance retention ratio of the asymmetric capacitor charged at 1.3 V as a function of the cycle number is shown in Figure 12. The impressive specific capacitance retention of 95% of the initial capacitance of our ASC after 2000 cycles demonstrated the greatly improved cycling stability [33]. We initially observed a small decrease in capacitance in the first 100 cycles, recognising that the electroactive material was not entirely accessible for ion diffusion. During cycling, inactive portions opened for the electrolyte ion diffusion of the working electrodes, which improved the capacitance retention and demonstrated the highest capacitance retention in the range of 100–2000 cycles [32,38]. The ASC showed nearly identical results and had better reversibility and rate capability, which were determined by inspecting the 2000 GCD cycles, as shown in Figure 12. We also tested the ASC cell’s columbic efficiency over 2000 cycles. The first cycle’s columbic efficiency was 99.9%, and the efficiency retained after 2000 cycles was 99.9%. Furthermore, the observed cycling stability and specific capacitance retention have real-time supercapacitor applications.

## 4. Discussion

In conclusion, we successfully synthesised the NS-2 (NiSn(OH)_6_) co-precipitation method and investigated its electrochemical capacitance in three- and two-electrode configurations. NS-2 exhibited superior electrochemical properties in terms of specific capacitance in a three-electrode system when compared with other samples. We created a hybrid supercapacitor with NS-2-positive and AC-negative electrodes, respectively. At an operating voltage of about 1.3 V in PVA/KOH gel electrolyte, the ASC demonstrated decent specific capacitance, comparatively high power and energy density, and reliable cycling stability. The coupling of NS-2 with AC to produce supercapacitors resulted in a high specific capacitance of 22.5 Fg^−1^ at a current density of 1 Ag^−1^. The organised optimisation of the positive and negative electrodes led to 5.85 Wh/kg energy density without sacrificing power density. Even after 2000 consecutive cycles, the GCD experiment demonstrated an admirable capacitance retention of 95%, which was a notable achievement of the ASC. From the intriguing results attained in our complete study on NS-2//AC ASC, we propose that NS-2 may be a suitable electrode to meet the increasing demand for high energy storage along with high-power density devices. The synthesis process can also be implemented to other double-transition metal hydroxides to improve capacitive performance.

## Figures and Tables

**Figure 1 nanomaterials-13-01523-f001:**
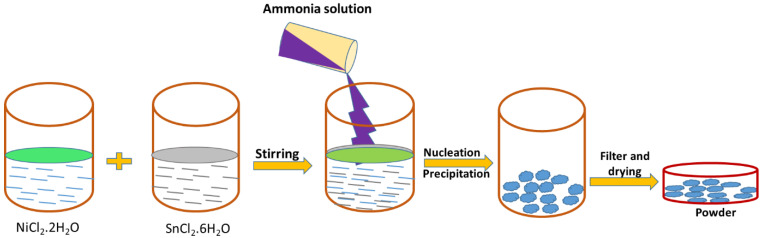
A schematic design for the precipitation technique of synthesising NiSn(OH)_6_ hydroxide nanoparticles.

**Figure 2 nanomaterials-13-01523-f002:**
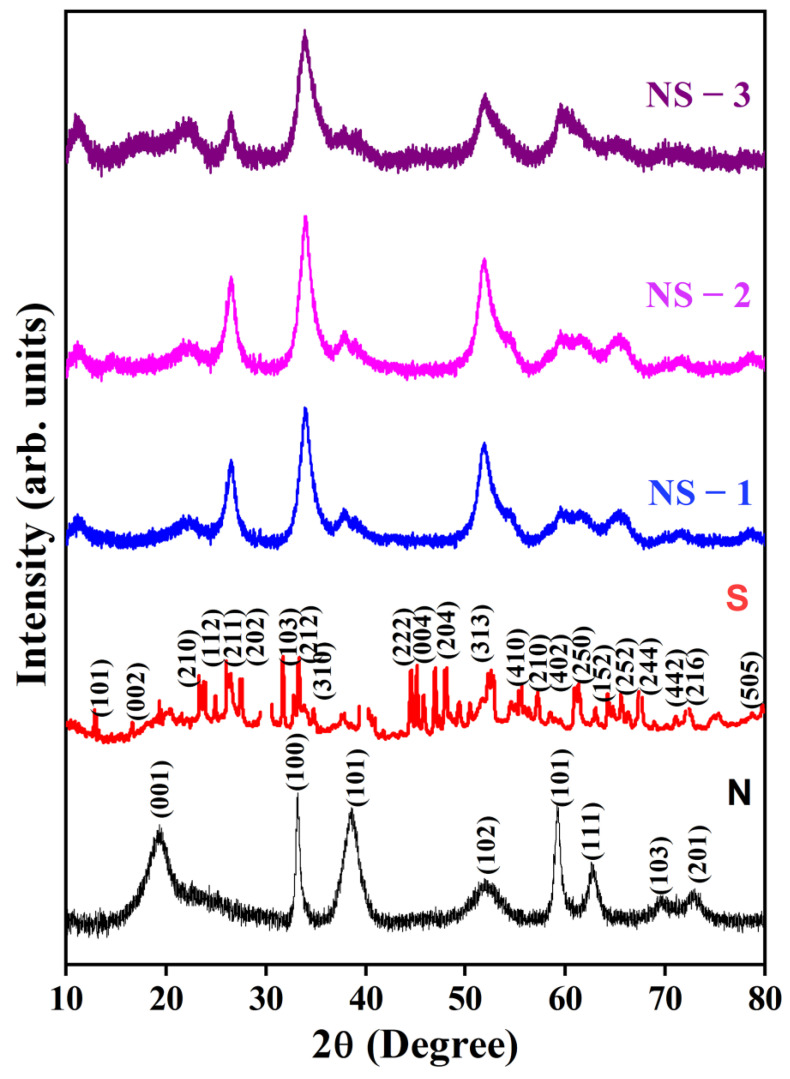
Powder X-ray diffraction (PXRD) patterns of NiSn(OH)_6_ hexahydroxide.

**Figure 3 nanomaterials-13-01523-f003:**
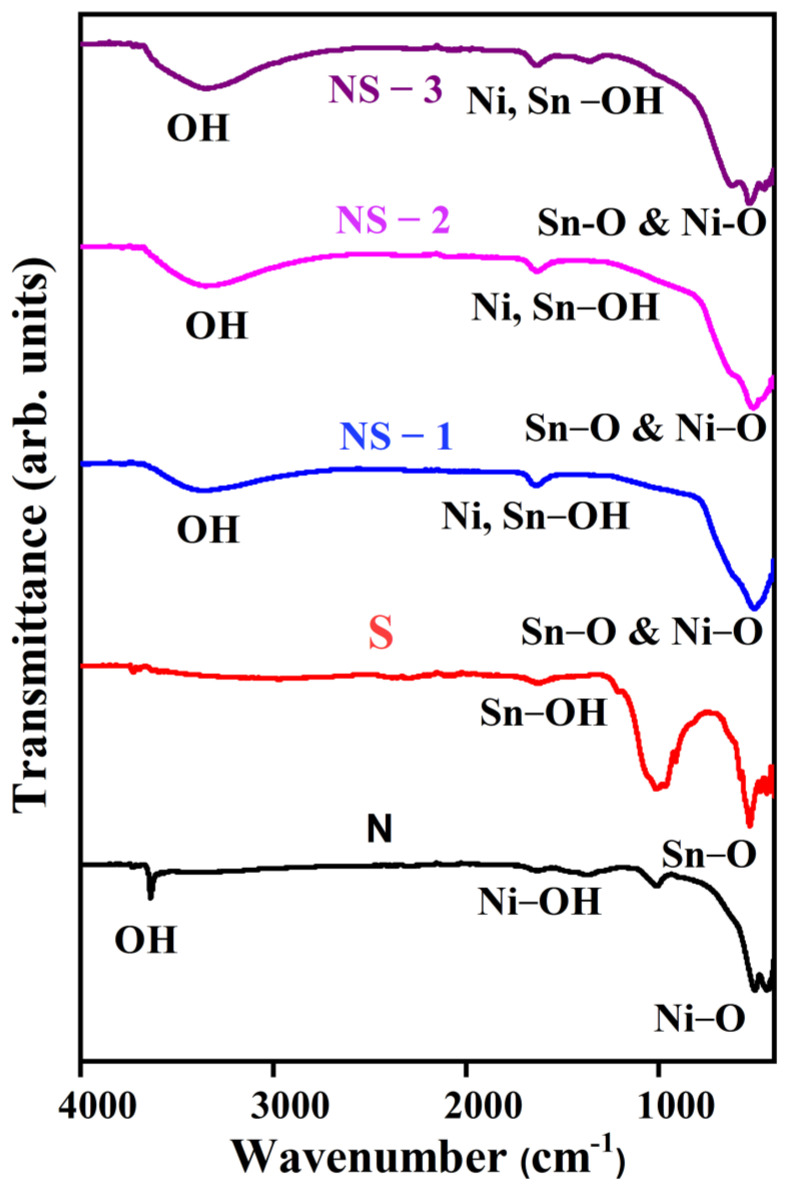
FT-IR spectra of Ni_x-y_Sn_y_(OH)_2_.

**Figure 4 nanomaterials-13-01523-f004:**
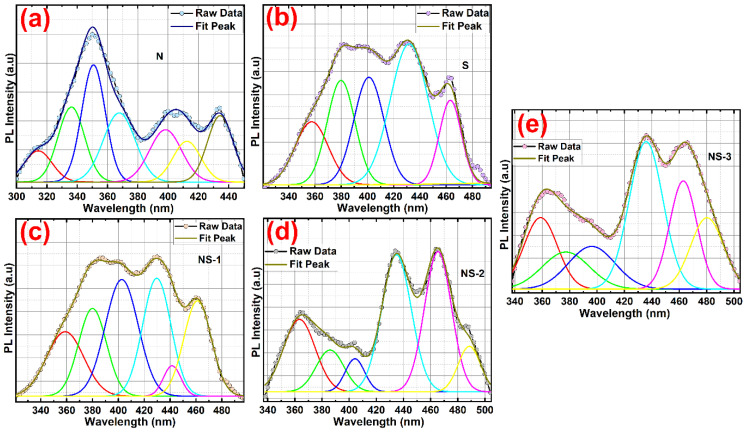
PL spectra and deconvoluted color line of (**a**) N sample (deconvoluted using 7 PL components, like red, green, navy blue and cyan color lines corresponds to UV emission, pink and yellow corresponds to violet—indigo emission and ash green line corresponds to blue emission) (**b**) S sample (deconvoluted using 6 PL components, like red and green corresponds to UV emission, navy blue and cyan corresponds to violet—blue emission, and pink line corresponds to green emission.), (**c**) NS—1 sample (deconvoluted using 6 PL components, like red and green lines corresponds to UV emission, blue and cyan lines corresponds to violet—indigo emission and pink and yellow lines corresponds to blue emission), (**d**) NS—2 sample (deconvoluted using 6 PL components, like red and green lines corresponds to UV emission, blue and cyan lines corresponds to violet—indigo emission and pink and yellow lines corresponds to blue—cyan emission) and (**e**) NS—3 sample (deconvoluted using 6 PL components, like red and green lines corresponds to UV emission, blue and cyan lines corresponds to violet—indigo emission and pink and yellow lines corresponds to blue—cyan emission).

**Figure 5 nanomaterials-13-01523-f005:**
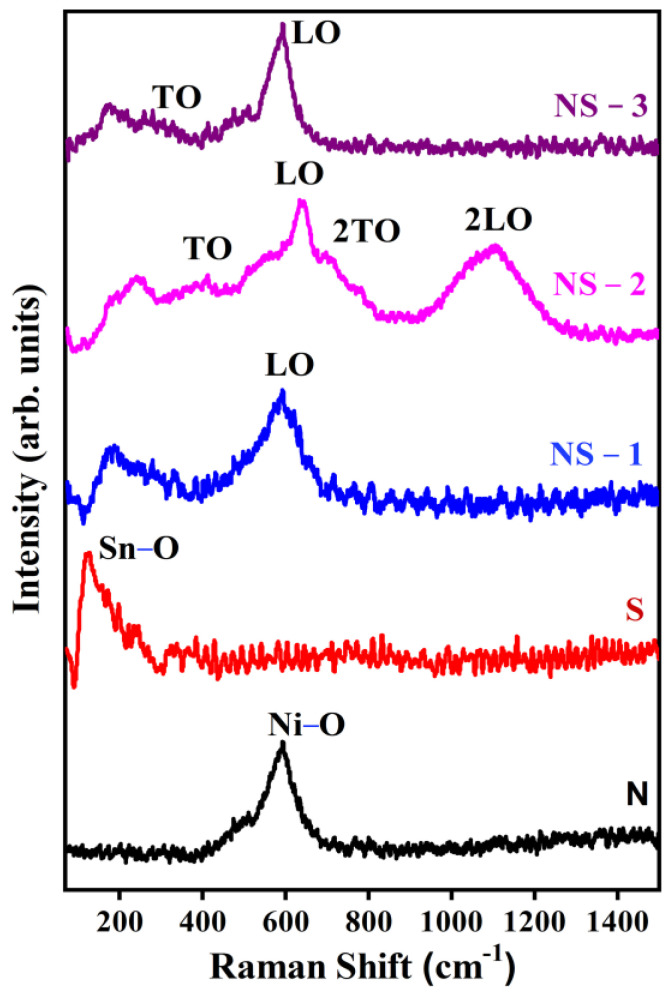
Raman spectra of Ni_x-y_Sn_y_(OH)_2_.

**Figure 6 nanomaterials-13-01523-f006:**
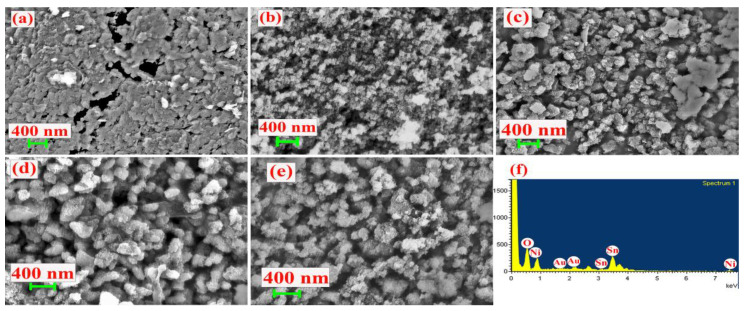
The morphology of (**a**) N, (**b**) S, (**c**) NS-1, (**d**) NS-2, and (**e**) NS-3 hydroxide nanoparticles; (**f**) EDAX spectra of NS-2 samples.

**Figure 7 nanomaterials-13-01523-f007:**
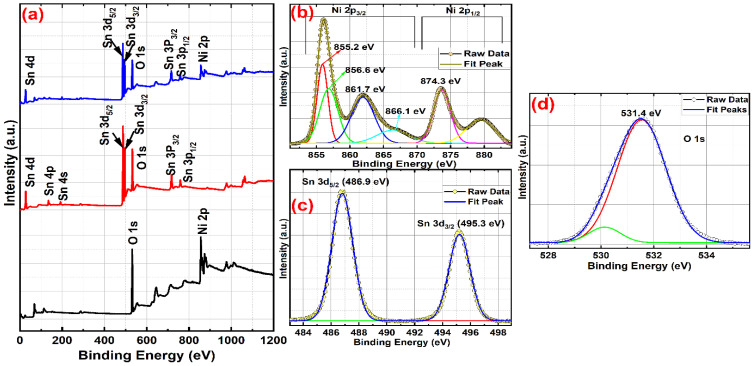
XPS survey scan (**a**) of (black) N, (red) Sn and (blue) SN-2. Detail XPS scan of (**b**) Ni 2p, split into two doublet spin state Ni 2p3/2 and Ni 2p1/2, these are further deconvoluted into red line represent Ni 2p3/2 satellite peaks, green line represent the primary peak which resembles the co-existence of Ni2+ vacancies and Ni3+ ions in Ni2+-OH, blue and cyan colour line are satellite peaks arise due to Ni 2p3/2 and Ni 2p1/2, rose line represent Ni3+ oxidation state and yellow line is the satellite arise due to Ni 2p1/2 (**c**) Sn 3d further splits into (red line hidden under cumulative fit peak) 3d5/2 and (green line hidden under cumulative fit peak) 3d3/2 and (**d**) O 1s is deconvoluted into two peaks, red line represents OH- ions and surface oxygen vacancy and green line represents Sn-O bonding for SnO_2_.

**Figure 8 nanomaterials-13-01523-f008:**
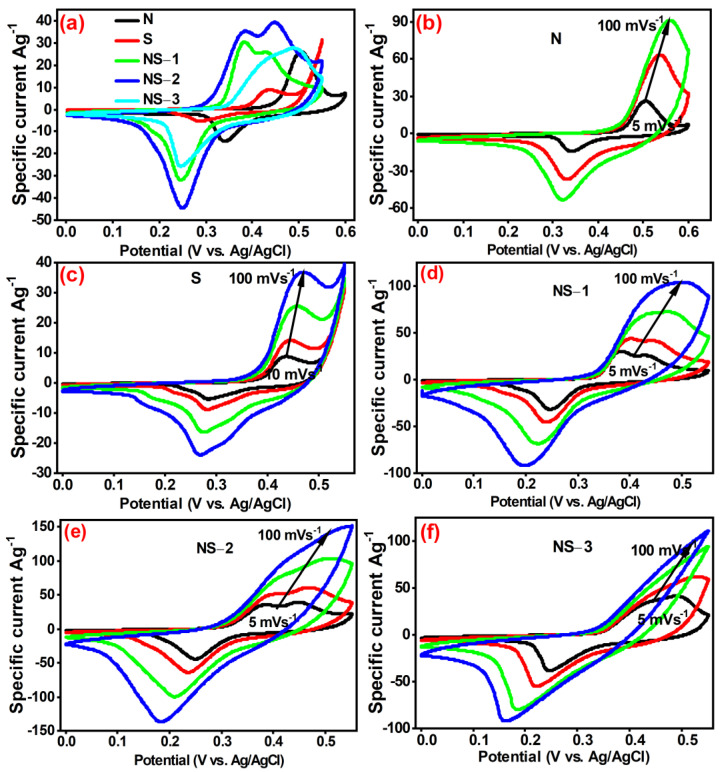
(**a**) CV curves of prepared samples at scan rate of 10 mVs^−1^ in 6 M KOH; CV plots of (**b**) N (Ni(OH)_2_), (**c**) S (Sn_4_O_4_(OH)_6_), (**d**) NS-1 (NiSn(OH)_6_), (**e**) NS-2 (NiSn(OH)_6_), and (**f**) NS-3 (NiSn(OH)_6_) with various scan rates.

**Figure 9 nanomaterials-13-01523-f009:**
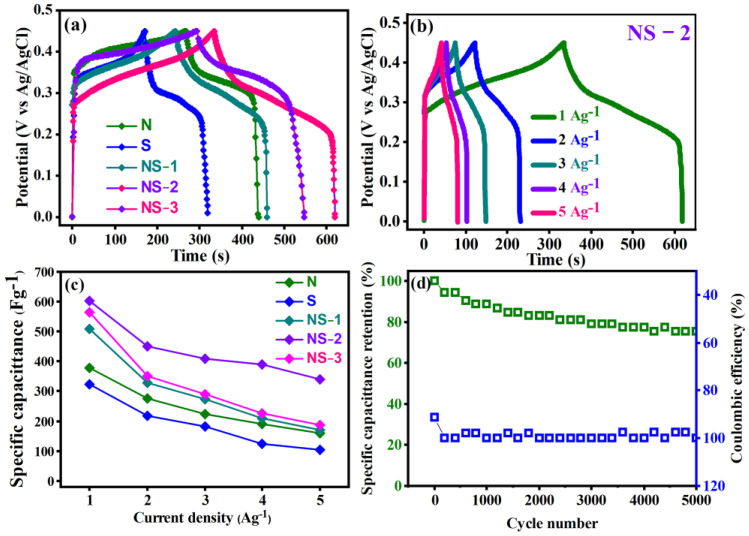
(**a**) GCD curves of prepared samples at current density of 1 Ag^−1^ in 6 M KOH, (**b**) GCD curves of NS-2 at various current densities (1–5 Ag^−1^), (**c**) specific capacitances of prepared samples calculated at different current densities, (**d**) long-term cycling stability and Coulombic efficiency of NS-2 measured at 10 Ag^−1^.

**Figure 10 nanomaterials-13-01523-f010:**
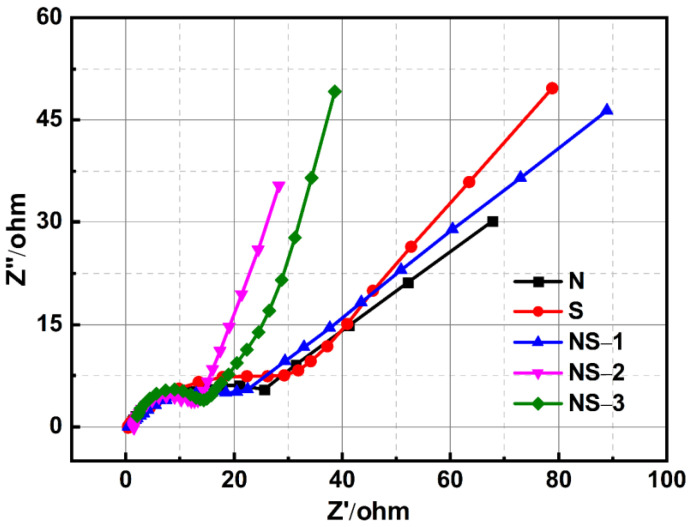
The EIS of Ni_x-y_Sn_y_(OH)_2_.

**Figure 11 nanomaterials-13-01523-f011:**
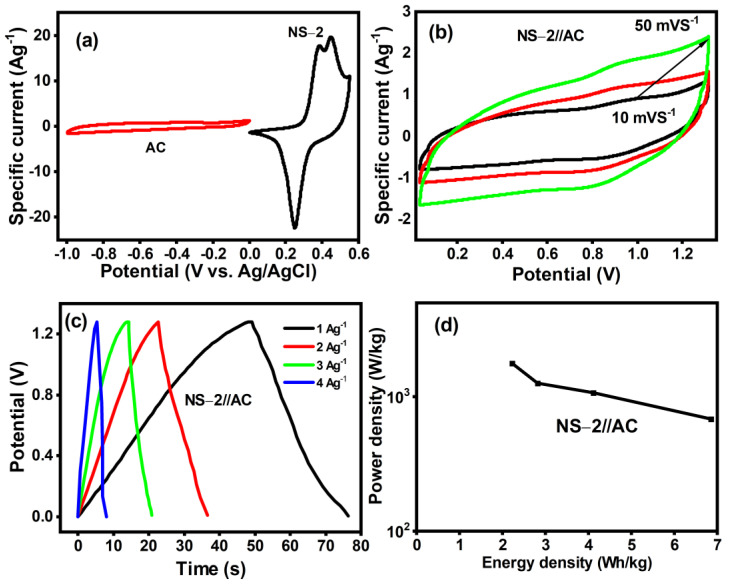
(**a**) Comparative CV curves of AC and NS-2 (NiSn(OH)_2_) performed using a three-electrode cell. (**b**) NS-2 (NiSn(OH)_2_)/AC ASC CVs at different scan rates. (**c**) CDs of ASC at different current densities. (**d**) Ragone plot of ASC.

**Figure 12 nanomaterials-13-01523-f012:**
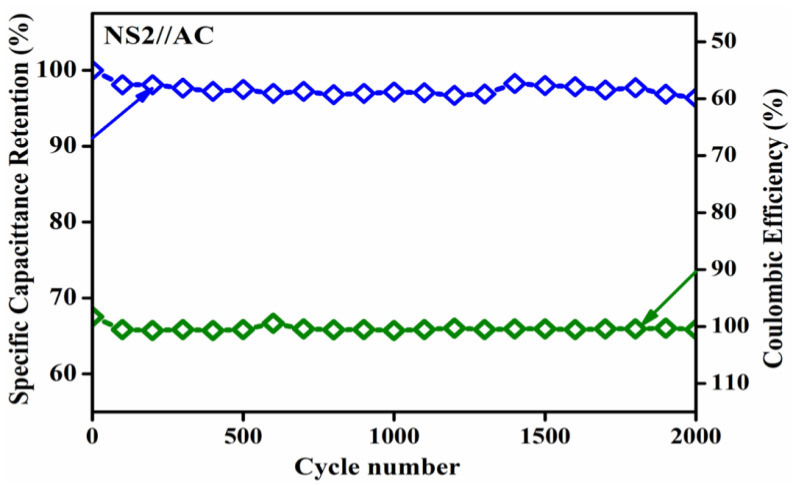
Cycling performance of ASC [Specific capacitance (Blue colour) and coulombic efficiency (Green colour)] at a current density of 5 Ag^−1^.

**Table 1 nanomaterials-13-01523-t001:** Element atomic percentage of prepared samples confirmed by EDAX.

Material	Ni (at.%)	Sn (at.%)	O (at.%)
N	52.46	---	47.54
S	---	34.66	65.34
NS-1	10.38	18.18	71.34
NS-2	13.83	13.68	72.49
NS-3	19.12	9.94	70.94

**Table 2 nanomaterials-13-01523-t002:** Specific capacitance of NiSn(OH)_6_ hydroxide nanomaterials.

Electrode Materials	Electrolyte	Specific Capacitance Fg^−1^ (Ag^−1^)	Capacitance Retention % (Ag^−1^)	Cycles	Ref.
CoAl LDH/graphene foam	6 m LiOH	101 (0.5)	100 (0.5)	5000	[32]
CoAl LDH/CNTs	2 m KOH	873 (0.5)	79.9 (1.72)	1000	[33]
NiCoFe-LDH/PANI	6 m KOH	408 (2)	84 (2)	1000	[34]
MgFe-LDH/PANI	2 m KOH	592.5 (2)	95 (2)	500	[35]
CoAl-LDH/rGO	6 m KOH	616.9 (1)	95.8 (10)	2000	[36]
LDH@P(NIPAM-co-SPMA)	2 m KOH	505 (1)	—	—	[37]
NS-2 (NiSn(OH)_6_)	6 m KOH	607	95.8 (5)	5000	This work

## Data Availability

Data and materials supporting the research are found within the manuscript. Raw data files will be provided by the corresponding author upon request.

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
