# Peer review of "Influence of Ni and Sn Perovskite NiSn(OH)6 Nanoparticles on Energy Storage Applications"

_nanomaterials, 2023, doi:10.3390/nano13091523_

Round 1
Reviewer 1 Report
1.Please provide TEM tests, including HRTEM, diffraction images, etc.
2.Please modify the format of the article carefully, the XPS pictures can not see.
3.Please supplement the growth mechanism of the sample.
4.The mechanism description of the property part is not detailed enough, please add it.
5.Please carefully supplement the synthetic part, the PH as neutral should not be 0.7.
Author Response
Response to Reviewer Comments
Manuscript ID: Nanomaterials-2342575
Journal: Nanomaterials-2342575
Dear Professor,
The authors are very much thankful to the reviewers for critically reviewing the manuscript and giving some suggestions which helped us to improve the manuscript. The necessary corrections are done according to the recommendations of the reviewers.

Reviewer 2 Report
This manuscript reported the synthesis of NiSn(OH)6 hexa-hydroxide nanoparticles with various concentrations of Ni2+ and Sn4+ ions for supercapacitor applications. The detailed structural and electrochemical properties of NiSn(OH)6 were investigated, and the electrochemical analysis indicated that NS-2 exhibited the highest capacitoance and cycling stability. Also, NS-2 based supercapacitor device delivered the largest energy density. Although NS-2 showed the impressive performance, the work can not be published in its present form. The main disadvantages were as follows:
1. Over the past years, lot of studies regarding layered double hydroxides (LDH) for supercapacitor electrodes have been published. Compared with previous researches, this work did not demonstrate significant advance in the preparation of nanoarchitecture or in the electrochemical performance of double metal hexahydroxide. In other words, the novelty was rather weak and it was somewhat like routine work.
2. Why was NiSn(OH)6 chosen for supercapacitor electrode? what was the advantages of NiSn(OH)6 over those LDH materials?
3. The authors attributed NS-2 with high performance to its more active sites? how to confirm this argument? A direct evidence should be given.
4. The discussion about electrochemical properties of NiSn(OH)6 electrodes was rather superficial. A more in-depth explanation should be provided.
5. EIS analysis should be presented to further investigate the electrochemical behavior of NiSn(OH)6 electrodes.
6. The usage of English language in this manuscript was quite unsatisfactory, please proofread it carefully.
Author Response

(The authors gave the same response as above.)

Reviewer 3 Report
The manuscript entitled “An influence of Ni and Sn hexahydroxide [NiSn(OH)6] nanoparticles for supercapacitor applic“ by G. Velmurugan et al. reports the applications of NiSn(OH)6/ activated carbon (AC) composites in the field of the asymmetric supercapacitor.
The following changes must to be carried out in the revised manuscript :
i) The sentence “The FTIR spectra obtained for N, S, NS-1, NS-2 and NS-3 [NiSn(OH)6] and are shown in Fig 3. “ should to be rewritten.
ii) A comment concerning UV emission detected at 379-395 nm needs to be included.
iii) The sentence “Oxidation and reduction peaks were observed at potential (0.35-0.5 V) and (0.35-0.25 V) scan rates of 10 mV/s, respectively. “ needs to be rewritten.
iv) The redox reaction shown on page 6 must be rewritten.
v) The ASC’s charge/discharge process shown on page 7 needs to be rewritten more clearly. Additionally, all reactions from this work must to be written considering stoichiometry.
vi) The deconvolution of the PL spectra needs to be revised, because at present this is not corrected. The authors should explain the number of the components used for this deconvolution.
vii) The authors should to explain the differences between Raman spectra of NS-2 versus NS-1 and NS-3.
viii) Figure 8 needs to be analysed more deeply.
I recommend this manuscript to be published in Nanomaterials only a major revision.
Author Response

(The authors gave the same response as above.)

Reviewer 4 Report
Recommendation: minor revision.
Comments: In this work, the authors synthesized NiSn(OH)6 containing double metal hexahydroxide via a co-precipitation process as an electrode for supercapacitors. The structure and physicochemical properties of as-prepared materials were studied in detail with suitable techniques and reasonably explained. The electrochemical capacitance of the NiSn(OH)6 was studied systematically with the effect of various concentrations of Ni2+ and Sn4+. The experiment data relevant to the NiSn(OH)6 offered in this manuscript are sufficient to support the conclusion. So, I recommend that this manuscript can be accepted for publication in Nanomaterials after minor revision.
1. The introduction of this paper needs to make a strong argument about the impact and novelty of the work further. So, the introduction should enrich some related articles in this section (such as Sensors and Actuators B: Chemical, 2017, 241: 298-307.; Carbon Neutralization, 2023, 2(1): 54-62.; Applied Surface Science, 2023, 611: 155672.).
2. All the Figures and Tables should offer the corresponding description and meet the requirement of the journal.
3. The XPS results displayed in Figure 7b are not accurate, because the fitted background shouldn't be a straight line.
4. The authors described in the manuscript that “with a 523 nm red laser wavelength” in the Experimental and synthesis techniques section. Maybe “with a 532 nm red laser wavelength”?
5. The authors better compare the electrochemical performance of current NiSn(OH)6 with reported related materials.
Author Response

(The authors gave the same response as above.)

Round 2
Reviewer 1 Report
The manuscript entitled "Effect of Ni and Sn hexahydrooxide [NiSn(OH)6] nanoparticles for supercapacitor applications" has been well revised. The authors have explicitly addressed the issues raised. Therefore, I propose that the current revised version of the manuscript can be published in Nanomaterials. Suggestion: accept.
Reviewer 2 Report
Since the issues raised by reviewers have been addressed, I recommend it to be accepted for publication in this journal.
Reviewer 3 Report
I recommend this paper to be published in Nanomaterials in the present state.